# Role of the Proportional Intake of Fortified Mother’s Own Milk in the Weight Gain Pattern of Their Very-Preterm-Born Infants

**DOI:** 10.3390/nu12061571

**Published:** 2020-05-28

**Authors:** Anna Petrova, Shannon Eccles, Rajeev Mehta

**Affiliations:** Department of Pediatrics, Rutgers Robert Wood Johnson Medical School, 89 French Street, New Brunswick, NJ 08901, USA; shanle@rwjms.rutgers.edu (S.E.); mehtara@rwjms.rutgers.edu (R.M.)

**Keywords:** preterm infants, fortified mother’s own milk, growth

## Abstract

Breastfeeding has been recommended for preterm infants as the optimal diet from nutritional, gastrointestinal, immunological, and developmental perspectives. However, the relevance of differing intakes of fortified mother’s own milk (MOM) on the growth of their preterm infants is a challenging question because of the potential risk of extrauterine growth impairment, apart from its essential role in the provision of biological and immunological factors, and the reduction of serious morbidities. We aimed to identify the weight gain pattern in very-preterm-born infants with respect to their proportional intake of fortified MOM. The daily and average weight gain, dietary volume, calories, and proportional intake of fortified MOM were studied in a cohort of 84 very-preterm-born infants during the first 2 weeks post initiation of full enteral feeds. Groups 1, 2, and 3 were comprised of infants with a proportional fortified MOM intake of 85% or more, 35% to 84.9%, and 0 to 34.9%, respectively. Data analysis included regression models and a group-based comparison of the number of infants with weight gain that would be considered minimally acceptable for normal intrauterine growth. The infants’ weight gain was not found to be associated with the proportional intake of fortified MOM or other feeding parameters. Overall, the intergroup variability in the proportion of infants with weight gain less than the lower limit of normal fetal growth was insignificant. During the first 2 weeks post initiation of full enteral feeds, the weight gain pattern of the studied very-preterm-born infants was not significantly dependent on the proportional intake of fortified maternal milk.

## 1. Introduction

Nutritional vulnerability due to inadequate nutrient stores at birth, and metabolic as well as gastrointestinal immaturity [1,2,3], are major factors that affect the growth and development of preterm infants not only during birth hospitalization but also during childhood and adult life [4,5,6,7]. The American Academy of Pediatrics (AAP) recommends that maternal milk should be the primary food for prematurely born infants [8] because of not only its nutritional but also its gastrointestinal, immunological, and developmental advantages [9,10,11]. However, the nutritional value of exclusive feeding with human milk is questioned despite the fortification that has been used for the achievement of appropriate protein uptake and the growth of very- or extremely-preterm-born infants [12]. A higher rate of short-term growth in formula-fed preterm neonates versus those fed donor human milk has been reported [13]. The study that included mothers who did not intend to provide milk to their extremely-preterm-born infants showed reduction of length gain but not weight in their infants randomized to receive fortified donor human milk, compared to those who were on bovine milk-based preterm formula [11]. However, a precise quantitative measure of the intake of maternal milk by the preterm infant is required for the identification of valid data regarding the relationship between breastfeeding and related outcomes [14]. The World Health Organization (WHO) suggests defining the proportional intake of mother’s own milk (MOM) as an indicator value for exclusive breastfeeding [15]. The proposed proportional intake of MOM for the definition of partial breastfeeding [16] is more attributable than objective. It has been suggested that the existing definitions of breastfeeding are unsuitable for breastfeeding research [17]. To our knowledge, there is no study that reports on the growth of very-preterm-born infants as a function of the proportional intake of fortified MOM. We believe that not only the exclusivity but also the relative intake of MOM should be assessed in order to provide recommendations for mothers with a reduced ability to support the breast milk feeding of their infants. In the present study, we determined the association between proportional intake of fortified MOM and weight gain in very-preterm-born infants. We hypothesized that because the intake of fortified MOM insures an adequate delivery of major nutrients, the proportion of MOM does not affect the weight gain in enterally fed very-preterm-born infants within the first 2 weeks of full enteral nutrition. The aim of the present study was to investigate the weight gain pattern of infants born at a gestational age of 32 weeks or less with respect to the proportional intake of fortified MOM during the first 2 weeks post initiation of full enteral feeding.

## 2. Methods

This report presents an analysis of clinical data collected as a part of a prospective observational cohort study that was approved by the Institutional Review Board (IRB) to investigate the energy metabolism mediators, type of feeding, and growth in very-preterm-born infants. The parents of all subjects gave informed consent before their babies were included in the study. The study was conducted in accordance with the Declaration of Helsinki, and the protocol was approved by the Institutional Review Board (IRB) of Rutgers Health Sciences on 1 April 2012 (IRB Protocol Number 0220120055). Human immunodeficiency virus (HIV)-negative mothers who delivered infants without congenital anomalies at a gestational age of 32 weeks or less were included in this study. Infants of consented mothers who had received full enteral feeds for at least 2 weeks prior to their discharge from the neonatal intensive care unit (NICU) were included in this report. We identified the infant’s weight gain pattern over the first 2 weeks of full enteral feedings with respect to the type of feeding that was defined as (i) predominantly breast fed if the average of the fortified MOM in the infant’s diet during the study period was 85% or more (Group1); (ii) partially breast fed if the average of the MOM intake was 35% to 84.9% (Group 2); and (iii) predominantly formula fed if the average of the MOM in the infant’s nutrition was 0% to 34.9% (Group 3).

As per the policy that existed at our center in 2012, preterm infants born at less than 32 weeks gestation whose birth weight was less than 1500 g received total parenteral nutrition (TPN). Infants with a gestational age of 32 weeks or less, but with a birth weight or 1500 g or more, were started on enteral feeds along with a supplemental infusion of dextrose, water, and electrolytes as required, for maintaining normal hydration and biochemical status. The collected research data included the duration of TPN, age at initiation of trophic feeds (days), chronological age (days) and postconceptual age (PCA) at which full enteral feeding was started (weeks), method of full enteral feeding, weight and PCA at discharge, and length of hospitalization. Additionally, we recorded maternal age; race/ethnicity; gravidity; parity; gestational age in completed weeks; type of pregnancy and mode of delivery; morbidities; medications; whether the Apgar score at 5 min was less or more than 7 [18]; and infant’s weight, length, and head circumference. Infants whose birth weight was below the 10th percentile on intrauterine growth curves were classified as small for gestational age (SGA) [19]. Morbidity data for the duration of the infant’s stay in neonatal intensive care unit (NICU) were also collected, and included sepsis, intraventricular hemorrhage (IVH), bronchopulmonary dysplasia (BPD), patent ductus arteriosus (PDA), and necrotizing enterocolitis (NEC).

### 2.1. Definition of Feeding Measures

For the purpose of this research, full enteral feeding was defined as tolerance by the baby of all the prescribed nutrition as milk feeds (either MOM or preterm formula) without any supplemental parenteral fluids or nutrition. The data that were collected during the study period included (i) the daily feeding volumes in milliliter, (ii) the proportion of MOM in the daily nutritional intake, (iii) the average of MOM intake in percent per day (percent/d), (iv) the average of the feeding volume in milliliters per kilogram per day (mL/kg/d), and (v) the daily and average caloric intake in kilocalories per kilogram per day (kcal/kg/d). While the study infants were on full enteral feeding, they were receiving 24 kcal/oz (80 kcal/100 mL) preterm formulas and/or fortified MOM wherein the fortification had been started when the overall enteral feeding volume reached 80 mL/kg/day. Our NICU uses standard fortification, which assumes that all breast milk has an average caloric content and macronutrient composition, and fortifies with a predetermined amount of fortifier. The nutritional value of the proportional intake of fortified MOM was calculated based on the reports that the average caloric content of human milk is 20 kcal/oz (66.7 kcal/100 mL) [20,21] and one pack of commercial powder fortifier (0.9 g) adds 4 kcal/oz (13.3 kcal/100 mL).

### 2.2. Definition of Growth Measures

Daily weights measured in grams were used to calculate the day-to-day and average weight gain in grams per kilogram per day (g/kg/d) during the first 2 weeks post initiation of full enteral feeds. We calculated the change in head circumference during the study period in centimeters per week (cm/wk) due to the high reliability of this growth parameter [22]. We found the weekly measured recumbent length data to be inaccurate for defining the linear growth, as also reported by Wood et al. [23], and hence did not include it in the final analysis.

### 2.3. Sample Size Calculation

The sample size was calculated to identify statistically significant inconsistency in weight gain (continuous variable) between the three groups using standardized effect size (Cohen *d* = 0.35). Cohen d is arbitrary, although it commonly refers to effect sizes as small (*d* = 0.2), medium (*d* = 0.5), and large (*d* = 0.80). We selected an effect size that was between small and medium to estimate the sample for ANOVA, which analyzed the statistical difference with respect to the within and between groups variability of weight gain in the studied subjects. A total of 76 subjects were required to show a minimum of 35% variability in the weight gain based on the type of feeding during the first 2 weeks post initiation of full enteral feeding with an alpha of 5% and power goal of 0.80. We decided to increase enrolment by 50% because it was difficult to predict the number of infants that might get discharged prior to the completion of at least 2 weeks of full enteral feeding.

### 2.4. Data Presentation and Statistical Analysis

We used chi-square and analysis of variance (ANOVA) followed by the Tukey test and nonparametric statistics (Kruskal-Wallis ANOVA) as required. Variables showing a difference between study groups at a level of *p* < 0.1 were entered into the multiple regression models to identify the independent effect of the proportional intake of MOM on the weight gain of the infants during the first 2 weeks post initiation of full enteral feeding. Additionally, we analyzed the significance of the maternal factors associated with the quantity of breast milk that the mother contributed for the infant’s diet. In accord with the AAP’s recommendations [20], we analyzed the infant’s weight gain with respect to the lower limit of the normal fetal growth of 14.4 g/kg/d [4] or 15 g/kg/d [5]. Data are reported as percentages (%), means, medians with interquartile ranges (25 and 75th percentiles), regression coefficient with Standard Error of estimate (β +/−SE), Odds Ratios (OR) with 95% Confidence Interval (95% CI), and correlation coefficients (r). STATISTICA 13.2 was used for the data analysis. All statistical tests were two-sided with the significance level set at *p*-value of less than 0.05.

## 3. Results

As shown in Figure 1, among the 84 mother-infant pairs included in the study, 37 (44.0%) were in Group 1, 16 (19.1%) in Group 2, and 31 (36.9%) in Group 3. In Group 1, the lower level of the MOM intake was 96.7%; in Group 2, the upper and lower levels varied between 55.4% and 69.9%; and in Group 3, the upper level of MOM intake was 9.0% (Figure 2). Among the Group 1 infants, 51.4% were fed exclusively MOM, whereas in Group 3, 35.5% of the infants were fed exclusively preterm formula. As compared to the mothers in Group 3, those in Groups 1 and 2 were more likely to be older, primiparous, and/or primigravidous, and either white or other than black/Hispanic (Table 1). In Group 1, maternal hypertension, lower gestational age, and birth weight of their infants were more frequently recorded than in Groups 2 and 3. The multivariate logistic regression models that were constructed revealed increased probability for establishment of predominant/partial breastfeeding after very-preterm-born deliveries by nulliparous mothers (OR 2.3, 95%CI 1.3, 4.0) and mothers of white/other than black and Hispanic race/ethnic backgrounds (OR 2.9, 95%CI 1.7, 5.0). As shown in Table 1, there was no difference in prematurity-related morbidities between the study groups.

### 3.1. Weight Gain during the First 2 Weeks Post Initiation of Full Enteral Feeds

The majority of the study participants received TPN with trophic feedings having been initiated at day 3–4 of life (Table 2). We found no significant variability between groups in chronological age (days) and PCA (weeks). The average intake (mL/kg) in Group 1 was slightly lower than in Group 3 (*p* = 0.05), which could be correlated with gestational age (*r* = 0.43, *p* < 0.0001). Infants in all three groups demonstrated comparable weight gain during the first 2 weeks post initiation of full enteral feeding (Figure 3). Regression analysis showed no association between the proportional intake of fortified MOM prior to (β 0.008+/−0.012, *p* = 0.5) and after controlling for relevant covariances such as gestational age (GA), birth weight (BW), PCA at initiation of full enteral feedings, and average daily intake (β 0.01+/−0.013, *p* = 0.46). None of the included covariances were independently associated with the infants’ weight gain pattern during the 2 weeks of observation: birth weight (β 0.003+/−0.003, *p* = 0.37), gestational age (β −0.239+/−0.5), PCA (β −0.069+/−0.38, *p* = 0.85), and enteral volume intake (β 0.036+/−0.064, *p* = 0.57). Additionally, the weight gain was lower than the fetal norm of greater than 14.4 g/kg/d [4] and 15 g/kg/d [5] in 37.8% (95%CI 22.9, 55.2) and 40.5% (95%CI 25.2, 57.8) of Group 1; 18.8% (95%CI 5.0, 46.3) and 31.3% (95%CI12.1, 58.5) of Group 2; and 38.7% (95%CI 22.4, 57.7) and 54.8% (95%CI 36.3, 72.1) of the Group 3 infants, respectively (*p* = 0.33–0.23). We did not find any significant difference in the head circumference gain (cm/week) between Group 1 (0.89, 95%CI 0.56, 1.22), Group 2 (0.71, 95%CI 0.3, 1.12), and Group 3 (1.13, 95%CI 0.72, 1.53), *p* = 0.39 in the first 2 weeks post initiation of full enteral feeds.

### 3.2. Duration of Hospitalization and Discharge Parameters

The duration of hospitalization varied between 67.2 days (95%CI 56.5–77.8) in Group 1, 48.8 days (95%CI 32.3, 65.4) in Group 2, and 51.2 days (95%CI 41.2, 61.1) in Group 3 (*p* < 0.05). However, the PCA (weeks) and weight (g) at discharge—37.4 weeks (95%CI 36.7, 38.0) and 2441 g (95%CI 2259, 2623) in Group 1, 37.6 weeks (95%CI 35.5, 39.7) and 2344 g (95%CI 2047, 2640) in Group 2, and 36.6 weeks (95%CI 36.0, 37.3) and 2354 g (95%CI 2216, 2491) in Group 3—were comparable (*p* = 0.29 and *p* = 0.64, respectively). Approximately 35% pf the infants were discharged between 0 and 7 days after being on full enteral feeds for 2 weeks, which included 18.9% of the infants in Group 1, 56.3% of those in Group 2, and 41.9% of the infants in Group 3 (*p* < 0.01).

## 4. Discussion

This study is the first that found no significant difference in the weight gain pattern of very-preterm-born infants during the first 2 weeks post initiation of full enteral feeds with respect to the proportional intake of fortified MOM and factors such as gestational age, postconceptual age (PCA), and feeding volumes. In about one-half of the studied very-preterm-born infants, the weight gain did not reach the lower limit of the fetal growth norm in a term pregnancy [4,5] irrespective of the proportion of fortified MOM in their diet. Although the effect of human milk fortification on the weight gain pattern of preterm infants is reportedly small [24], one possible explanation for our findings is that the fortification of MOM may have enabled achievement of a caloric intake that was similar to preterm formula. It is difficult to compare our results with studies that have looked only at the efficacy of donor human milk in growth of premature infants versus those on preterm formula. Some have reported decreased in-hospital growth [13] and higher rates of discharge with weight below the 10th percentile in donor breast milk-fed very low birth weight (VLBW) infants, compared to those fed preterm formula [4,25]. Nevertheless, Chowning et al. [26] have reported a lack of association between the discharge weight of VLBW infants and the number of days on any (unidentified) quantity of donor human milk in their diet. Various aspects of the pasteurization associated differences between the MOM and donor milk, including the protein [27,28] lipid and oligosaccharide content, and other biological properties [29] should be taken into consideration when utilizing results obtained from breast feeding research.

We would like to acknowledge the limitation of the observational design of our study. However, a randomized trial would not be appropriate because the AAP recommends maternal milk as the first choice for the preterm infants’ diet [8]. Additionally, the conclusion regarding the weight gain pattern is restricted by the observation that lasted only 2 weeks. The measurements were confined to the 2-week period to avoid any observational bias due to the variability between the study groups in the remaining duration of the hospitalization after the completion of 2 weeks of full enteral feedings. 

## 5. Conclusions

We found no association between the proportional intake of fortified maternal milk and the weight gain pattern in very-preterm-born infants during the first 2 weeks post initiation of full enteral feeding. The use of fortified mother’s own milk may not place very-preterm-born infants at an increased risk for growth restriction. If so, any amount of maternal milk that very-preterm-born infants consume would be beneficial due to the contained biological and immunological factors. However, our conclusions are necessarily limited due to the short period of observation. Further studies will be required to identify the association between the proportional intake of maternal milk and infants’ future health and development.

## Figures and Tables

**Figure 1 nutrients-12-01571-f001:**
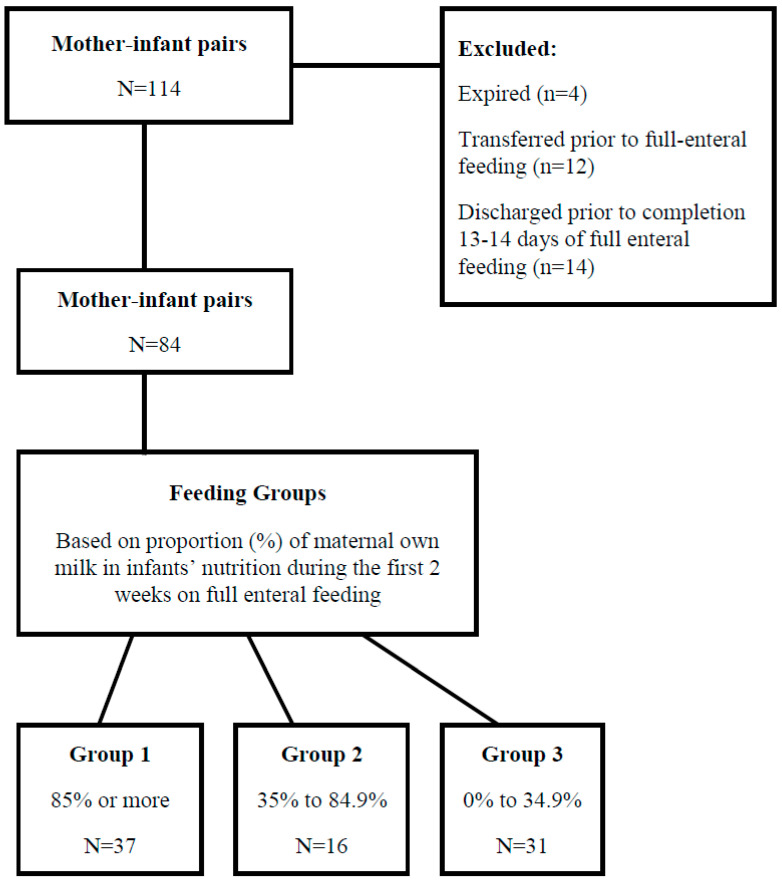
Cohort enrollment diagram and inclusion–exclusion criteria.

**Figure 2 nutrients-12-01571-f002:**
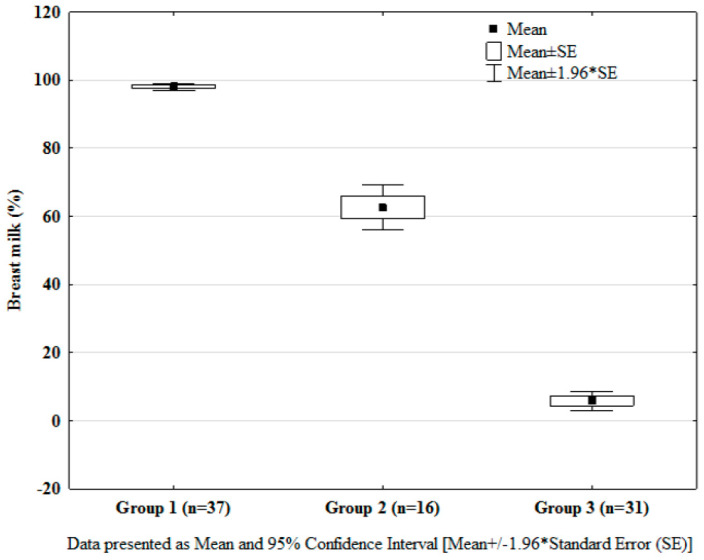
Proportion of mother’s own milk in the full enteral feeds of the studied infants.

**Figure 3 nutrients-12-01571-f003:**
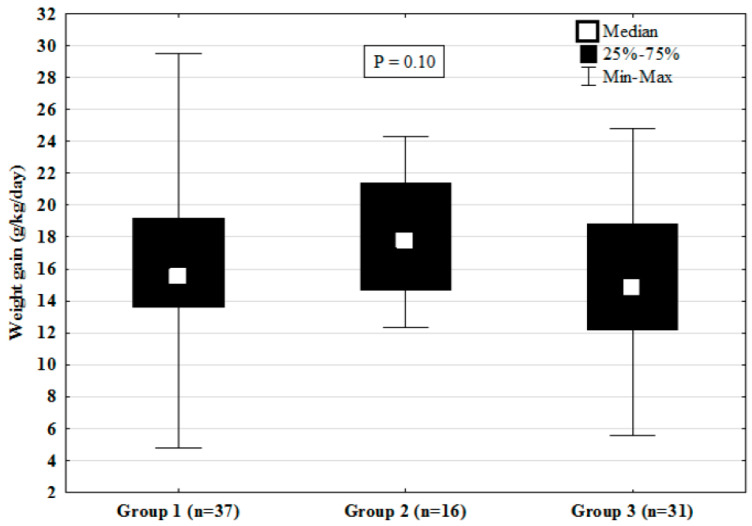
Group based comparison of median and interquartile ranges (IQR) of weight gain (g/kg/d) during the study period.

**Table 1 nutrients-12-01571-t001:** Maternal and neonatal characteristics based on the type of enteral feeding *.

Characteristics	Feeding Groups			*p* Value
	Group 1 (*n* = 37)	Group 2 (*n* = 16)	Group 3 (*n* = 31)	
Maternal Age, y *	32.2 (30.7, 33.8)	35.8 (32.0, 39.5)	30.5 (28.1, 32.9)	<0.02
Race/ethnicity, *n* (%)	*n* = 32	*n* = 14	*n* = 28	<0.01
White	14 (43.8)	7 (50.0)	8 (28.6)	
Black	6 (18.8)	0	13 (46.4)	
Hispanic	0	2 (14.3)	5 (17.9)	
Other	12 (37.4)	5 (35.7)	2 (7.1)	
Primipara, *n* (%)	20 (54.1)	12 (75.0)	7 (22.6)	<0.01
Primigravida, *n* (%)	12/34 (35.3)	8 (50.0)	3 (9.7)	<0.04
Singleton, *n* (%)	15/36 (41.7)	9 (56.3)	22/30 (70.0)	0.07
Cesarean section, *n* (%)	30/36 (83.3)	11 (68.8)	25 (80.7)	0.44
Morbidity, *n* (%)				
PIH,/pre-eclampsia ^a^	5/36 (13.9)	8 (50.0)	13 (41.9)	<0.01
Diabetes	1/36 (2.8)	1 (6.3)	3 (9.7)	0.26
Intrapartum, *n* (%)				
Steroids	21 (58.3)	12 (75.0)	24 (80.0)	0.14
Antibiotics	12 (32.4)	5 (31.3)	17 (54.8)	0.13
Magnesium Sulfate	6 (16.2)	2 (12.5)	8 (25.8)	0.58
Male gender, *n* (%)	19 (51.4)	11 (68.8)	13 (41.9)	0.18
Gestational age, week *	27.7 (26.8, 28.6)	29.8 (28.6, 31.0)	29.1 (28.2, 30.3)	<0.02
Birth weight, g *	1027 (924, 1321)	1285 (1130, 1439)	1272 (1102, 1442)	<0.02
Birth length, cm *	35.8 (34.6, 37.1)	38.4 (36.9, 39.9)	37.9 (36.4, 39.3)	<0.03
Head circumference, cm*	25.4 (24.5, 26.3)	27.0 (26.0, 28.0)	26.5 (25.3, 27.7)	0.102
SGA ^b^, *n*, %	4 (10.8)	2 (12.5)	3 (9.7)	0.95
5 min Apgar <7, *n* (%)	7 (18.9)	1 (6.3)	9 (29.0)	0.17
Sepsis, *n* (%)	15 (40.5)	6 (37.5)	12 (38.7)	0.95
IVH, *n* (%) ^c^	9 (24.3)	4 (25.0)	6 (19.4)	0.85
BPD, *n* (%) ^d^	7 (18.9)	4 (25.0)	4 (19.4)	0.89
PDA, *n* (%) ^e^	18 (48.6)	4 (25.0)	8 (25.8)	0.09
NEC, *n* (%) ^f^	2 (5.4)	0 (0)	1 (32)	0.60

* Continuous data presented as mean (95%CI); ^a^ PIH, Pregnancy-Induced Hypertension; ^b^ SGA, Small for Gestational Age; ^c^ IVH, Intraventricular Hemorrhage; ^d^ BPD, Bronchopulmonary Dysplasia; ^e^ PDA, Patent Ductus Arteriosus; ^f^ NEC, Necrotizing Enterocolitis.

**Table 2 nutrients-12-01571-t002:** Comparison of feeding parameters based on the type of feeding.

Characteristics	Feeding Groups			*p* Value
	Group 1 (*n* = 37)	Group 2 (*n* = 16)	Group 3 (*n* = 31)	
Received TPN, *n* (%)	26 (70.3)	12 (75)	17 (58.6)	0.14
PPA ^a^ of trophic feeding initiation *	3.2 (2.5, 3.9)	3.6 (0.2, 6.9)	3.6 (2.2, 5.1)	0.91
PPA ^a^ to full enteral feed *	21.6 (17.5, 25.8)	17.4 (11.3, 33.0)	17.4 (12.3, 22.6)	0.25
PCA ^b^ to full enteral feed *	30.8 (30.1, 31.5)	32.2 (31.5, 33.0)	31.5 (30.7, 32.3)	0.06
Enteral volume intake (mL/kg/d) *	138.2 (135.7, 141.9)	143.3 (138.1, 148.6)	145.9 (140.7, 151.0)	0.05
Caloric intake (kcal/kg/d) *	113.4 (110.6, 116.2)	114.1 (109.7, 118.4)	115.1 (114.6, 118.7)	0.86

^a^ PPA, postpartum age (days); ^b^ PCA, postconceptual age (weeks); * continuous data presented as mean (95%CI).

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
