# Peer review of "Role of the Proportional Intake of Fortified Mother’s Own Milk in the Weight Gain Pattern of Their Very-Preterm-Born Infants"

_nutrients, 2020, doi:10.3390/nu12061571_

Round 1

Reviewer 1 Report

I carefully reviewed the paper entitled “Role of the Proportional Intake of Fortified Mother’s Own Milk in the Weight Gain Pattern of their Very Preterm Born Infants”. Surely this is a topic of great interest and there is still a great debate in literature on the best way to feed very preterm infants. However this study presents some important limitations mainly in the methodology that could influence the results. The main limit is the lack of data regarding the NICU staying and the development of prematurity-related complications in the 3 study groups. This reviewer suggests to add these data to make stronger your findings.

MAJOR

1) INTRODUCTION, page 2: Please, better clarify the aim of the study. It is not clear to this reviewer what the basis is for the study hypothesis.

2) METHODS, page 2: How did you choose the percentages for groups’ definitions? Please, clarify.

3) METHODS, page 2: Please, clarify is you fortified all the meals from the 80 ml/kg/die.

4) METHODS, page 3, Sample size calculation: Please, add the reference that you used to calculate the sample size.

5) TABLE 1, page 5: Why the number of pregnancy (as for PIH/preeclampsia, diabetes, CS) has been calculated for 34 subjects in the Group I?

6) TABLE 1, page 5: Gestational age and birth weight were significantly lower in Group I; this could influence results.

7) TABLE 2, page 6: Only the 70% in Group I and the 58% in Group III received TPN although extreme prematurity. Please, specify in the methods section the policy at your center for beginning TPN in preterm infants.

8) RESULTS, page 5, Weight gain during the first 2 weeks post initiation of full enteral feeds: “Regression analysis showed persistence of these findings even after controlling for relevant co-variances such as gestational age, birth weight, PCA at initiation of full enteral feedings, and average daily intake (data not presented)”. Please, add these data.

9) RESULTS, page 5, Weight gain during the first 2 weeks post initiation of full enteral feeds: “None of the included co-variances were independently associated with the infants’ weight gain pattern during the 2 weeks of observation”. Please, specify in the methods section which were the co-variances. It is not clear.

10) RESULTS, page 6, Duration of hospitalization and discharge parameters: Please, add HC data at discharge.

11) Did you considered prematurity-associated complications such as PDA, RDS, BPD etc… These could influence results. How many infants after birth were supported with nCPAP, how many were mechanically ventilated etc…? The lack of this information is a major limit of this study thus complications experienced after birth could have a great influence on growth.

MINOR

12) METHODS, page 2: Please, add the definition of full enteral feeding used for this study.

13) METHODS, page 2, Definition of feeding measures: Please, transform all oz in ml. It is not clear for international readers.

14) FIGURE 1, page 4: Please, improve the quality of this figure.

15) TABLE 1, page 5: It is not clear the significance. The p is for a trend?

16) RESULTS, page 5, Weight gain during the first 2 weeks post initiation of full enteral feeds: “We found no significant variability between groups in chronological age (days) and PCA (weeks)”. Do you refer to full enteral feeding? Please, specify.

Reviewer 2 Report

General: In this prospective observational study, the authors monitored weight gain among very preterm infants within 3 cohorts defined by the proportion of intake that consisted of maternal breast milk. The manuscript is generally well written and incrementally advances our understanding of neonatal nutrition options, but enthusiasm is tempered by inclusion of a single center with a limited sample size.

Major Comments

  1. Abstract conclusion: you did not find significant differences, but that does not mean they were not present; consider replacing “was found to be independent of” with “was not significantly dependent on”.
  2. Methods: there appeared to be nice separation of proportion of MOM intake between the cohorts, but what was the rational used to define the cohorts (<35% vs. 35-85% vs >85%)?
  3. Discussion: consistent with my 1st comment, consider changing “no difference” to “no significant difference” in the 1st sentence.
  4. Conclusion: would change “does not” to “may not” and “hence” to “if so”

Minor Comments

  1. Grammar is strong, but there are a few minor issues such as “affects” rather than “affect” in the first sentence of the introduction.
  2. Table 1 has a high percentage of “other” race/ethnicity. Does that include women that did not self-identify or was there a specific race/ethnicity that could be further delineated?
  3. Was donor breast milk not available or not offered as an alternative to formula?

Round 2

Reviewer 1 Report

I carefully reviewed the revised version of the paper entitled “Role of the Proportional Intake of Fortified Mother’s Own Milk in the Weight Gain Pattern of their Very Preterm Born Infants”. Authors correctly answered this reviewer’s questions and the manuscript has been accordingly improved. I have only few minor revision.

MINOR

1) INTRODUCTION, page 2: Please, put in full the first time you cite (etc: MOM).

2) METHODS, page 2: Please, put the aim of the study at the end of the introduction section.

Author Response

Dear Reviewer 1,

Thank you for reviewing the manuscript entitled "Role of Proportional Intake of Fortified Mother's Own Milk in the Weight Gain of Very Preterm Born Infants" and suggesting two minor revisions. 

  1. Introduction, page 2: Please put in full the first time you cite (etc: MOM).  Response: This has been done in blue font in the corrected manuscript.
  2. Methods, page 2: Please put the aim of the study at the end of the introduction section. Response: This has been done in blue font in the corrected manuscript. Thank you.
